# GUBS: Graph-Based Unsupervised Brain Segmentation in MRI Images

**DOI:** 10.3390/jimaging8100262

**Published:** 2022-09-27

**Authors:** Simeon Mayala, Ida Herdlevær, Jonas Bull Haugsøen, Shamundeeswari Anandan, Nello Blaser, Sonia Gavasso, Morten Brun

**Affiliations:** 1Department of Mathematics, University of Bergen, 5020 Bergen, Norway; 2Department of Clinical Medicine, University of Bergen, 5021 Bergen, Norway; 3Neuro-SysMed, Department of Neurology, Haukeland University Hospital, 5053 Bergen, Norway; 4Department of Informatics, University of Bergen, 5020 Bergen, Norway

**Keywords:** brain tissues, non-brain tissues, segmentation, minimum spanning tree

## Abstract

Brain segmentation in magnetic resonance imaging (MRI) images is the process of isolating the brain from non-brain tissues to simplify the further analysis, such as detecting pathology or calculating volumes. This paper proposes a Graph-based Unsupervised Brain Segmentation (GUBS) that processes 3D MRI images and segments them into brain, non-brain tissues, and backgrounds. GUBS first constructs an adjacency graph from a preprocessed MRI image, weights it by the difference between voxel intensities, and computes its minimum spanning tree (MST). It then uses domain knowledge about the different regions of MRIs to sample representative points from the brain, non-brain, and background regions of the MRI image. The adjacency graph nodes corresponding to sampled points in each region are identified and used as the terminal nodes for paths connecting the regions in the MST. GUBS then computes a subgraph of the MST by first removing the longest edge of the path connecting the terminal nodes in the brain and other regions, followed by removing the longest edge of the path connecting non-brain and background regions. This process results in three labeled, connected components, whose labels are used to segment the brain, non-brain tissues, and the background. GUBS was tested by segmenting 3D T1 weighted MRI images from three publicly available data sets. GUBS shows comparable results to the state-of-the-art methods in terms of performance. However, many competing methods rely on having labeled data available for training. Labeling is a time-intensive and costly process, and a big advantage of GUBS is that it does not require labels.

## 1. Introduction

The brain is a complex organ that makes the central nervous system together with the spinal cord. It is divided into forebrain (sensory processing, higher reasoning), midbrain (motor movement, audio/visual processing), and hindbrain (autonomic functions such as sleep and respiratory rhythms). Over the past years, non-invasive imaging techniques have gained momentum, in assessing brain injury and studying brain anatomy. In particular, magnetic resonance technology is widely used in the diagnosis of brain diseases such as brain tumors, multiple sclerosis, hematomas and to find the cause of conditions such as epilepsy and headaches [1]. The technology produces magnetic resonance imaging (MRI) data, which can be processed to produce 3D volumetric data with the intensity of voxels varying according to the properties of different tissues. MRI images are most commonly presented as a stack of two-dimensional slices. Analysis of such high quality complex MRI data is a tedious process. Recent advances with computer aided-tools have overcome the major pitfalls of manual MRI data analysis. Brain MRI segmentation is an important step in image processing as it highly influences the outcome of the entire analysis, which is crucial in the case of surgical planning, delineating lesions and image-guided interventions. To segment any target structure in the brain, it is common to perform a preprocessing step to isolate the brain from non-brain tissues such as the skull, dura and scalp [2,3].

Methods for segmenting the brain from non-brain tissues can be classified as manual, semi-automated, and fully automated [4,5]. Manual brain extraction gives high precision but is labor-intensive [6]. Semi-automated methods involve a certain degree of user intervention, whereas automated methods do not depend on any human interaction. Most automated methods for brain extraction can be classified in categories, such as mathematical morphology-based, intensity-based, deformable surface-based, atlas-based, and hybrid methods [2,7,8]. Machine learning techniques, including neural networks are also widely used for skull stripping [9,10]. We review different skull stripping techniques before describing GUBS.

Louis Lemieux et al. [11] proposed a fully automated method for segmenting the whole brain in T1- weighted volume. It is fast and based on foreground thresholding and morphological operations. It performs 3D connected component analysis at each level. Furthermore, Brain Extraction Tool (BET) is an automated method for segmenting Magnetic Resonance head images to separate brain and non-brain tissues. It uses a deformable model that evolves to fit the brain’s surface by applying a set of locally adaptive model forces. The method is robust, fast and it does not require any pre-registration or preprocessing [12].

FreeSurfer is an open-source software that implements various image processing tools for both structural and functional MRI data sets. Skull stripping is one of the tasks that FreeSurfer provides for processing MRI images. It performs automatic skull stripping from intensity normalized images through a deformation of a tessellated ellipsoidal template into the shape of the inner surface of the skull [13]. David W. Shattuck and Richard M. Leahy [14] present an MRI analysis tool that produces cortical surface representations with spherical topology from MR images of the human brain. One of the tools that include skull stripping is a brain surface extractor (BSE). It breaks connections between the brain and non-brain tissues by using a morphological erosion operation, and then it identifies the brain using a connected component operation followed by a dilation operation to undo the effects of the erosion [2]. The final step is closing small holes that may occur in the brain surface.

The work of [15] presents a skull-stripping algorithm based on a hybrid approach (HWA). It combines watershed algorithms and deformable surface models. The method uses a localized voxel in T1-weighted white matter to create a global minimum in the white matter. The global minimum is created before applying the watershed algorithm with a pre-flooding height. The Robust Brain Extraction (ROBEX) is another method for performing skull stripping. It is a learning-based brain extraction system that combines discriminative and generative models. It is trained to detect brain boundaries using the random forest approach and ensures that the result is plausible. To obtain the final segmentation, the imperfect shape presented by the model is refined by using a graph cut [16]. The work presented in [17] proposed an automatic skull stripping method based on deformable models and histogram analysis named Simplex Mesh and Histogram Analysis Skull Stripping (SMHASS) method. It defines the starting point for deformation using a rough segmentation based on thresholds and morphological operators. The deformable model is based on a simplex mesh, whereas its deformation is controlled by local gray levels of the image and the information obtained on the gray level modeling of the rough-segmentation.

A simple skull stripping algorithm (S3) is proposed in [18]. The method is based on brain anatomy and image intensity characteristics. It is a knowledge-based algorithm and works by using adaptive intensity thresholding and then morphological operations. Oeslle Lucena et al. [9] proposed a Convolutional Neuron Network (CNN) for brain extraction in MRI trained with “silver standard” masks. The method generates silver standard masks which are used as inputs by using the Simultaneous Truth and Performance Level Estimation (STAPLE) algorithm and then implementing a tri-planar method using parallel 2D U-Net-based CNNs, named as CONSNet. Anam Fatima et al. [19] proposed a skull stripping method and evaluated it on a 2D slice-based multi-view U-Net (MVU-Net) architecture. It performs as well as a 3D model while using fewer computational resources. In [20], a Single-Input Multi-Output U-Net (SIMOU-Net) was developed for segmenting foetal brain. Different from the original U-Net, the SIMOU-Net has a deeper architecture and takes account of the features extracted from each side output. Furthermore, Ref. [6] proposed a 3D-UNet for skull stripping to address the entire brain extraction problem satisfactory for diverse datasets. The work in [21] proposed a graph-based method for skull stripping, which uses intensity thresholding on the input image to obtain a preliminary mask. Finally, the method removes narrow connections using graph cuts followed by post-processing.

In this paper, we propose a graph-based approach that represents an MRI image using a graph, before segmenting the brain. The method uses the minimum spanning tree (MST) of the constructed adjacency graph to separate the brain, non-brain tissues, and the background. The approach involves the following steps; preprocessing of the MRI image, sampling points within and outside the brain, MST construction from the graph, isolating brain, non-brain tissues, and the background, followed by post-processing. The general idea of using a graph-based method for skull stripping relates to the ideas of [21]. We define an image into adjacency graph in the same way, but our approach differs significantly regarding the segmentation criteria. We perform skull stripping using the MST of the graph, in which each vertex has a minimal number of connections instead of using the whole graph. Different from our approach, the method presented by [21] depends on an estimate of the region bounded by white matter obtained by region growth. We define edge weights differently, and our approach does not depend on region growth. The method presented by [21] uses an initial mask obtained by thresholding and uses a graph cut to disconnect the narrow connections. GUBS depends on the MST constructed from the adjacency graph and separates the brain, non-brain tissues, and the background by disconnecting paths connecting these regions.

## 2. Materials and Methods

### 2.1. Materials

In this paper, we analyzed three different data sets. The first data set is from the Open Access Series of Imaging Studies (OASIS). It consist of T1W images from 77 subjects with an isotropic voxel resolution of 1.0 mm and a shape of 176×208×176 [22]. The ground truth segmentation provided with this data set was created using a custom method based on registration to an atlas and then revised by human experts. Twenty of the subjects in this dataset suffered from Alzheimer’s disease, a degenerative disease characterized by the loss of brain tissue [9,16].

The second data set consists of 20 simulated T1W MRI images mimicing healthy brains collected from the BrainWeb (BW) database [23,24]. They are provided as *.MINC* format anatomical models consisting of a set of 3D tissue membership volumes, one for each tissue class. Each label at a voxel in the anatomical model represents the tissue that contributes the most to that voxel. They have dimensions of 362×434×362 and have a 0.5 mm isotropic voxel size.

The third data set consists of T1W MRI images from 18 subjects provided to the Internet Brain Segmentation Repository (IBSR) by the Center for Morphometric Analysis at Massachusetts General Hospital [25]. The images are stored in the NIfTI format. The shape of the images is 256×256×128 and the voxel resolution is 0.94 mm × 0.94 mm × 1.5 mm [26]. The database contains a manual segmentation of the gray matter (GM), white matter (WM) and cerebral spinal fluid (CSF), as well as skull stripped images.

Figure 1 shows coronal sections of brains sampled from a 3D MRI of a representative subject from each of the three data sets.

### 2.2. Methods

In this section, a graph-based method for brain segmentation using MST is presented. An adjacency graph is constructed from a preprocessed 3D MRI and then its MST is applied for separating the brain, non-brain tissues, and the background. The summarized steps for brain segmentation are presented in Figure 2.

Let *A* be an MRI image with dimension M×N×L such that $A_{\left(i,j,k\right)}$ gives the intensity value at position (i,j,k) for i=0,1,⋯,M−1; j=0,1,⋯,N−1 and k=0,1,⋯,L−1. Voxels at position (i,j,k) and $\left(i^{\prime },j^{\prime },k^{\prime }\right)$ are called adjacent if ${\left(i-i^{\prime }\right)}^{2}+{\left(j-j^{\prime }\right)}^{2}+{\left(k-k^{\prime }\right)}^{2}=1$. We can mathematically define
(1)$$A=\left[A_{\left(i,j,k\right)}\right]\in R^{M\times N\times L}$$
and each point changes along the coordinate axes with respect to the position of the adjacent voxels.

#### 2.2.1. Preprocessing

We read all the image data sets using a *NiBabel* package that can read different common medical images file formats [27] and then retrieve a 3D image using a *nilearn* package [28]. Finally, every MRI image is convert into .npy using a *numpy* package [29]. Each image was resized to the size of 128 × 128 × 128. The intensities were scaled in the range of 0 to 1 using
(2)$$A_{\left(i,j,k\right)}^{scaled}=\frac{A_{\left(i,j,k\right)}-A_{\left(j,k\right)}^{\min}}{A_{\left(j,k\right)}^{\max}-A_{\left(j,k\right)}^{\min}},$$
where $A_{\left(j,k\right)}^{\min}=\min_{i\in \{0,\cdots ,M-1\}}\left\{A_{\left(i,j,k\right)}\right\}$ and $A_{\left(j,k\right)}^{\max}=\max_{i\in \{0,\cdots ,M-1\}}\left\{A_{\left(i,j,k\right)}\right\}$. We adapt the formula and its implementation from scikit-learn [28]. Since the implementation requires only number of samples and number of features, the 3D MRI are reshaped into 2D before scaling and then reshape back to 3D after scaling. Whenever $A_{\left(j,k\right)}^{\max}$ is approximately or equal to $A_{\left(j,k\right)}^{\min}$ the situation is taken care by setting scales of near constant features to 1 to avoid division by very small number or zero values.

We remove noise and make sure that there is no spot in the background (outside the head) by removing small objects. Some data sets may need additional preprocessing steps to accelerate the separation of the regions. Since the brain is connected to the rest of the head by dark, thin segments [11], a small threshold may be applied to disconnect more the brain from the non-brain tissues. This is done by setting all values less than the threshold to 0. For IBSR and OASIS data sets, the values 0.25 and 0.32 were used in the preprocessing step, respectively. For data sets with good contrast between the brain and non-brain tissues such as BW, thresholding is unnecessary.

#### 2.2.2. Edge Surface Detection

The edge surfaces in 3D images are defined as the structural boundaries of objects in the image. However, the true edge-surfaces of 3D images are continuous surfaces rather than discrete 3D edges like points which are detected by edge detectors [30]. These are points sampled at the surface that form the boundary between the object and the background. In this context the surface of the object is represented by intensity changes in the data volume [31]. The changes will be detected between edges.

The edges are detected by highlighting the local variation between the adjacent voxels. Example of the two adjacent voxels (i,j,k) and $\left(i^{\prime },j^{\prime },k^{\prime }\right)$ located on the opposite side of an edge. We use the finite difference method (particularly forward difference) to compute the change between two adjacent voxels from the binary image. We define *B* to be the binary image of *A*. The change is computed by
(3)$$\varepsilon =B_{\left(i^{\prime },j^{\prime },k^{\prime }\right)}-B_{\left(i,j,k\right)}.$$

For a 3D binary image the edge is detected when the changes between two neighboring voxels are
(4)$$\varepsilon =\left\{\begin{array}{cc} +1 & if\;B_{\left(i^{\prime },j^{\prime },k^{\prime }\right)}=1\;and\;B_{\left(i,j,k\right)}=0\\ -1 & if\;B_{\left(i^{\prime },j^{\prime },k^{\prime }\right)}=0\;and\;B_{\left(i,j,k\right)}=1 \end{array}\right.$$

So, when ε is +1 or −1 the forward difference method asserts the presence of an edge. The edge surfaces of the 3D image consist of all points satisfying the changes in Equation (Equation 4) along a given axis. Then, the discrete 3D edge-like points represents the set of the edge points detected from 3D binary image *B* by satisfying the condition ε for the neighboring voxels.

In the implementation we perform two steps to obtain the discrete 3D edge-like points. Image *A* is binarized to obtain image *B* by either setting voxels to 1 if the gray value is greater than zero or setting to 0 elsewhere. We fill all possible holes in the binary image *B* by using a function for binary closing from Scipy package [32]. Then, we use a *diff* function from numpy package [29] to compute the discrete difference of the binary image by using the forward difference formula. Then, the points satisfying condition ε are discrete 3D edge-like points distributed at the surface of the head. These points are very useful in the next subsection when sampling points within the brain and non-brain tissues.

#### 2.2.3. Sampling Points within the Brain, Non-Brain Tissues and Background

Let *T* be the set of points in the MRI image *A*, i.e.,
(5)T={(i,j,k):0≤i≤M−1,0≤j≤N−1,0≤k≤L−1}.

Next, we will sample points $T_{B}$, $T_{NB}$ and $T_{BG}$ in the brain, non-brain tissues and background, respectively. We define, $T_{B}\subset T$, $T_{NB}\subset T$ and $T_{BG}\subset T$ such that $T_{B}$, $T_{NB}$ and $T_{BG}$ are mutually exclusive but not exhaustive. That is,
(6)$$T_{B}\cap T_{NB}=T_{NB}\cap T_{BG}=T_{B}\cap T_{BG}=\emptyset$$
and
(7)$$T_{B}\cup T_{NB}\cup T_{BG}⊊T.$$

We do the following things to obtain points in $T_{B}$: From the discrete 3D edge-like points distributed at the surface of the head we take points above the selection line which is obtained by visual inspection (see Figure 3). Then, pull these points from the surface of the head to the brain by a certain distance $\delta_{1}\in N$. Then, compute the mean point of the pulled points. For each pulled point a distance is computed from the mean point and get the maximum distance. Then, the maximum value times 0.75 is the a distance threshold value for removing the points with the distance above the threshold value. This last step is performed to restrict the pulled points to lie within the brain and remove points close to the skull.

To obtain points $T_{NB}$ in the non-brain tissues we use all the discrete 3D edge-like points distributed at the surface of the head and pull them inside towards the non-brain tissue by $\delta_{2}\in N$. The pulled points are rechecked if there is any point with the intensity value of zero it is removed. See Figure 4 and Figure 5.

To obtain points $T_{BG}$ sampled in the background we use the binary image after closing all the possible holes (this is the same binary image described in the last paragraph of Section 2.2.2).

First, we compute all points in the binary image whose points have an intensity values of 0. We also find points at the six surfaces of the 3D cube of the binary image whose points have an intensity values of 0. We combine these points to increase the chance of getting representative points from all the sides. We then sample 20,000 of these points uniformly at random to get the $T_{BG}$ points.

#### 2.2.4. Graph

Let (V,E) be the voxel adjacency graph constructed from *A* such that *V* is the set of nodes and *E* represents a set of weighted edges. Each node represents a voxel at location (i,j,k) in *A*, and each edge connects two adjacent voxels. The edge’s weight is the absolute value of the difference between the intensity values of the voxels it connects.

#### 2.2.5. Collapsing Nodes

Given a graph *G* and a subset *H* of the nodes in *G* we construct a graph where all the nodes in *H* have been collapsed to a single node. In order to ease the construction of this graph we represent the single collapsed node by a node h∈H.

The graph $G_{H}^{h}$ where all nodes in *H* have been collapsed to a single node *h* in *H* is constructed as follows:

For every edge *e* in *G*, if both end nodes of *e* appear in *H* the edge *e* is discarded.For every remaining edge *e*, if a node *v* in *H* appears in *e*, the edge *e* is modified by replacing *v* by *h*.Remove all the nodes in *H* except the node *h* from the modified graph to obtain the graph $G_{H}^{h}$.

#### 2.2.6. Segmentation Criteria

The central problem addressed by GUBS is to segment an image using samples of nodes from the regions of interest. From the voxels adjacency graph, we construct a new graph by successively collapsing the nodes sampled from each of the regions. The main idea of GUBS is to use the minimum spanning tree of this new graph instead of the minimum spanning tree of the voxels adjacency graph.

Given an MRI image of the brain, we want to separate the voxel adjacency graph into three regions representing the brain, non-brain tissues, and background.

Let $V_{B}\subset V$, $V_{NB}\subset V$ and $V_{BG}\subset V$ be sets of nodes in a graph *G* corresponding to sets of coordinate points $T_{B}$, $T_{NB}$ and $T_{BG}$ sampled from brain, non-brain tissues, and background, respectively. We construct the graphs $G_{B}$, $G_{NB}$ and $G_{BG}$ so that the nodes $V_{B}$, $V_{NB}$ and $V_{BG}$ are in $G_{B}$, $G_{NB}$ and $G_{BG}$, respectively.

Let $v_{B}\in V_{B}$, $v_{NB}\in V_{NB}$ and $v_{BG}\in V_{BG}$ be randomly sampled nodes. Let $A=G_{V_{B}}^{v_{B}}$, $B=A_{V_{NB}}^{v_{NB}}$, $C=B_{V_{BG}}^{v_{BG}}$. In the modified graph *C* each of the sets $V_{B}$, $V_{NB}$ and $V_{BG}$ have been collapsed to a single node.

The minimum spanning tree (MST) of the graph *C* is the spanning tree with the least total edge weight among all possible spanning trees of the adjacency graph [33].

A path *P* in MST is a sequence of nodes in which each pair of consecutive nodes are connected by an edge. Note that since MST is a tree every pair of nodes is connected by a unique path.

The MST is constructed from the modified graph C and the components representing brain, non-brain tissues and the background are extracted as follows: First, we modify the MST by removing the edge with highest edge weight in the path connecting the nodes $v_{B}$ and $v_{NB}$ to separate the brain ($G_{B}$) and non-brain ($G_{NB}$) subtrees. Next, we separate the non-brain region and the background by removing the edge with highest edge weight in the path connecting the nodes $v_{NB}$ and $v_{BG}$. This gives us the connected components of the tree with three different labels. Finally, the labels are reshaped back to the shape of the input MRI image.

## 3. Results

For visualization purposes, we perform a padding of background slices on the MRI image. The visualization of the results in this paper are performed using Matplotlib [34] and Seaborn [35]. The segmented 3D brain from OASIS, BW and IBSR data sets are visualized in Figure 6, Figure 7 and Figure 8, respectively.

Figure 6, Figure 7 and Figure 8 represent visual comparison of the extracted 3D brain using the GUBS approach to the gold standard segmented brain. The general quality of the predicted brain is sufficient, except for small variation in finer details. The small variation can be attributed to different reasons such as the quality of the data (see Figure 1), differences when preparing the gold standard segmentation, the patients conditions, such as disease or aging, and possibly parameter tuning for certain data.

Figure 9 and Figure 10 present examples of a visual comparison of 2D slices segmented from two different subjects from IBSR and OASIS data sets, respectively. Notice that for Figure 9, the original sagittal sections show that some parts of the 3D head were cut. Figure 10 also present representative sagittal sections from a single subject. Since the proposed approach depends on sampling points within different regions, tuning the parameters for extracting the brain from the two compared MRI volumes (Figure 9 and Figure 10) can differ significantly.

### 3.1. Performance Analysis

In this subsection, we present a quantitative analysis using Jaccard Index (JI), Dice Similarity Coefficient (DSC), sensitivity, specificity, accuracy and precision to assess the performance of GUBS. We evaluate the performance by computing the listed measures of similarity between the predicted and the ground truth of the MRI image of the binary labels.

The voxels in the binary labels that are correctly classified as brain are represented as true positive (TP). The voxels that are incorrectly classified as brain are represented as false positive (FP). The voxels that are correctly classified as non-brain tissues are represented by true negative (TN). The voxels that are in the brain region but incorrectly classified as non-brain tissues are represented by false negative (FN). The Jaccard Index and Dice Similarity Coefficient are computed by
(8)$$JI=\frac{TP}{TP+FP+FN},\;DSC=\frac{2TP}{2TP+FP+FN}.$$

The sensitivity and specificity which show the percentage of brain and non-brain voxels are computed by
(9)$$Sensitivity=\frac{TP}{TP+FN},\;Specificity=\frac{TN}{TN+FP}.$$

The precision gives the ratio of the correctly positive identified labels against all the labels, whereas accuracy gives the ratio of the correctly identified.
(10)$$Precion=\frac{TP}{TP+FP},\;Accuracy=\frac{TP+TN}{TP+FP+FN+TN}$$

The values for the measures of similarity are presented in the range of 0 to 1. A measure of similarity with 0 value shows that there is no overlap between the predicted brain and the ground truth. The measure of similarity with 1 indicates that there is a perfect overlap between ground truth and the predicted brain segmentation.

The quantitative evaluation of GUBS compared to the state of the art method is presented in Table 1, Table 2 and Table 3 for OASIS, BW and IBSR data sets, respectively. The general performance of GUBS is satisfactory. For some data sets GUBS does not outperform other state of the art methods, but the results obtained by GUBS is consistent with previous results. Looking at the performance from the individual data sets GUBS does not perform well on sensitivity for OASIS and IBSR, but it performs well on specificity from all the three data sets.

#### Consistency Analysis of the GUBS across Different Data Sets

We have segmented three different data sets from OASIS, BW, and IBSR. They are significantly different in terms of quality and quantity. The OASIS data sets were collected from participants of different age groups, healthy and unhelthy subjects suffering from dementia. The data sets from BW are normal simulated brains whereas the IBSR data sets were collected from healthy subjects. To better understand the performance of the GUBS for differences in brain shapes across ages, possibly diseased and non-diseased brains, we combined all the results to explore the relationship between different measures of similarity. Figure 11 presents plots in matrix format showing the relationship between different measures of similarity on the three data sets. The diagonal subplots show the distributions for a specific measure of similarity in the individual data sets.

From Figure 11, the distribution in the diagonal show that the accuracy, DSC, and sensitivity from the OASIS data set have a lower deviation from the mean value compared to BW and IBSR data sets. It also indicates that specificity and precision from BW data have a lower deviation from the mean compared to the specificity and precision from OASIS and IBSR. Furthermore, Figure 11 shows that BW has the highest mean values for all measures of similarity compared to OASIS and IBSR data sets. Measures of similarity from IBSR show a higher deviation from the mean values compared to the measures from other data sets.

The pair plots show that the DSC are positively correlated with the accuracies obtained from all the data sets. The sensitivity obtained from BW and IBSR is positively correlated with accuracy and DSC. Specificity and precision obtained from BW show that they are not correlated with other measures of similarity whereas specificity and precision obtained from IBSR and OASIS are negatively correlated with sensitivity obtained from these data sets. For all data sets the obtained precision is positively correlated with the specificity.

Figure 12d shows a separation between the measures with and without outliers after combining the results from the three data sets. JI shows many outliers compared to accuracy and DSC. Based on the results presented in Table 1, Table 2 and Table 3 it is possible that the noted outliers come from BW results. Furthermore, we note that sensitivity, specificity, and precision do not have outliers even though specificity shows high performance for all the data sets compared to sensitivity and precision.

Looking at the size of the box plots we note that the measures of accuracy, DSCs, and specificity in their distribution show that the lower and upper quartile are close to each other compared to other measures.

### 3.2. Parameter Selection

Due to high variation within and across image data sets, it is challenging to get a single set of parameters that works to produce the best possible segmentation results for different image data sets. In this work, parameter tuning was initially done on trial and error bases to obtain a value that could achieve good results. For each data set, the GUBS is run by testing different parameter values and choosing the parameters which lead to the separation of the tree into the separated components. Figure 13 and Figure 14 show experimentation for different threshold parameters with same nodes sample size of 20,000, where Figure 15 and Figure 16 show experimentation for different nodes sample sizes and same threshold. Note that BW data (see Figure 16) does not need threshold.

The selection line, $\delta_{1}$ and $\delta_{2}$ can highly vary based on the size of the images. The selection of parameters for the selection line depends on the visual inspection because the 3D MRI undergo different preprocessing steps after acquisition of the signals. For OASIS, IBSR and BW data sets the selection line values used were 60, 65 and 35, respectively. For all the three data sets $\delta_{1}=15$. For IBSR and OASIS $\delta_{2}=1$, for BW $\delta_{2}=3$.

Figure 9 and Figure 10 show experimental result of one subject from IBSR and OASIS, respectively. The challenging part is to sample within the non-brain tissues because some parts of the brain and non-brain tissues have been removed (see row one in Figure 9). Some points sampled from the non-brain tissues are likely to be taken from the brain when $\delta_{2}$ is applied. For Figure 9 and Figure 10, $\delta_{1}=15$ and $\delta_{2}=1$.

### 3.3. Experimental Timing

The time complexity analysis for implementing GUBS is divided into three parts. These include adjacency graph construction, MST construction, and the time is taken for disconnecting the MST into different connected components. The adjacency graph size is defined by the number of voxels in the MRI image, corresponding to vertices in the adjacency graph. The experimental timing for the adjacency graph and MST construction is efficient and presented in [36]. For the adjacency graph constructed from MRI image 134 × 134 × 134 (with 2,406,104 nodes in the adjacency graph), we update the MST twice to disconnect a path connecting the brain and the non-brain tissues as well as disconnect the path connecting non-brain tissues and the background. The experimental time ranges between 12 and 25 s for disconnecting a path. The time spent on separating the tree into different connected components depends on the length of the path between the terminal nodes. The implementation was done by writing scripts in the python programming language, and it was run on a PC processor (Core i7-8650UCPU @ 1.90GHz×8).

## 4. Discussion

We extended the segmentation criteria used in the paper [36] by collapsing subgraphs of the voxels adjacency graph before constructing the MST and presented the GUBS method for segmenting the brain from MRI images. GUBS works by representing MRI volume into an adjacency graph, followed by collapsing representative nodes sampled from the brain, no-brain, and background regions, respectively. Then, the MST is constructed from the modified graph. The collapsed nodes are used as terminal nodes for disconnecting the paths in the MST to separate the brain, non-brain tissues, and background.

The approach was tested by segmenting the brain from three different MRI data sets. The results are compared to the ground truth to assess the performance of GUBS. The experimental results show that GUBS successfully segments brain with high performance from different data sets regardless of the differences of these data sets. Moreover, the results obtained using GUBS are compared to the results obtained using different state of the art methods. GUBS gives competitive results in terms of performance. However, unlike different state of the art methods which require labeled images for training, GUBS does not require labels. In most cases the labeled images are obtained manually by highly qualified individuals. This task is labour intensive and time consuming. That is one of strength of GUBS.

The evaluation of the results obtained using the GUBS approach from the three data sets indicates that the quality of the data highly influences the results. The results obtained from the BrainWeb data set indicate that GUBS outperforms different state of the art methods, whereas results obtained from the OASIS data set provide competitive results compared to the ones obtained previously using neural network approaches. In some cases, the specificity obtained by the GUBS approach outperforms other methods. Similar to other methods, GUBS get good results from IBSR data set. Since the approach depends on sampling points within different regions, the quality of the data might have influenced the IBSR results because some parts of the MRI images were removed. It is likely that nodes sampled in the wrong region will be collapsed together with nodes in a wrong region. Thus, the GUBS approach will be limited in the setting of incomplete 3D brain MRIs. An extension of GUBS to a semi-automated method, in which experts can sample points with high control before running the approach would alleviate this problem.

## 5. Conclusions

We developed a graph-based method that uses a minimum spanning tree (MST) to segment 3D brain from MRI images. The proposed method was tested by segmenting three different data sets. It is efficient and competes the state-of-the-art methods in terms of performance. The proposed method is simple to adapt and apply on different data sets.

## Figures and Tables

**Figure 1 jimaging-08-00262-f001:**
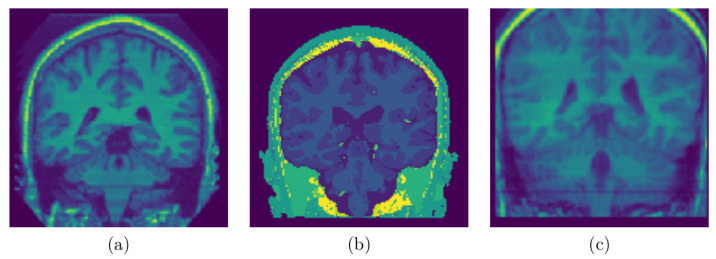
Sampled Slices: (**a**) Sampled slice coronal section from a 3D MRI volume OASIS data set, (**b**) Sampled slice coronal section from a 3D MRI volume BW data set, (**c**) Sampled slice coronal section from a 3D MRI volume IBSR data set. Notice the differences and the quality of data sets.

**Figure 2 jimaging-08-00262-f002:**
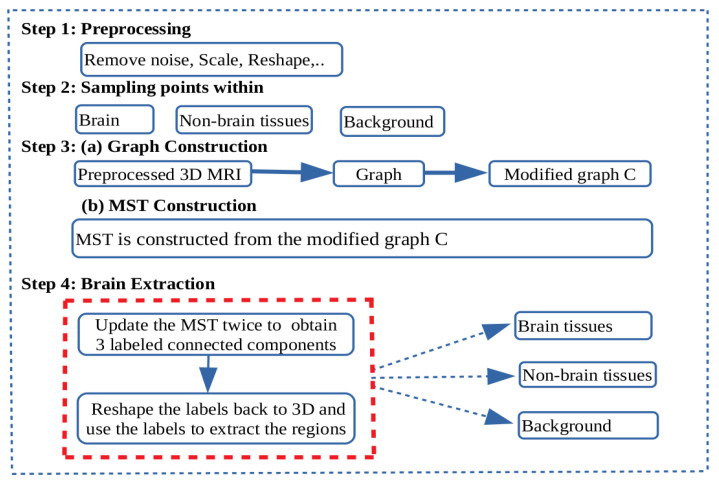
Schematic diagram: Flow diagram showing the steps for brain extraction. Step 1: Pre-processing to remove noise, scale values in the range of 0 and 1 and reshape the 3D magnetic resonance imaging (MRI) volume. Step 2: Sampling points within the brain, non-brain tissues, and the background. Step 3: An adjacency graph weighted by absolute intensity differences is constructed from the preprocessed 3D MRI volume. Then, nodes in the adjacency graph corresponding to the sampled points in step 2 are collapsed in their respective regions to form a graph *C*. From the modified graph *C*, a minimum spanning tree is constructed. Step 4: Brain segmentation. Nodes in *C* representing each of the regions of interest are the terminal nodes for the paths to be disconnected to separate the regions. First, the minimum spanning tree (MST) is modified by removing the edge with highest edge weight in the path connecting the representative nodes to separate the brain and non-brain subtrees. Again the MST is modified by removing the edge with highest edge weight in the path connecting the representative nodes to separate the non-brain and background subtrees. New labels are assigned and reshaped back to the shape of the 3D MRI.

**Figure 3 jimaging-08-00262-f003:**
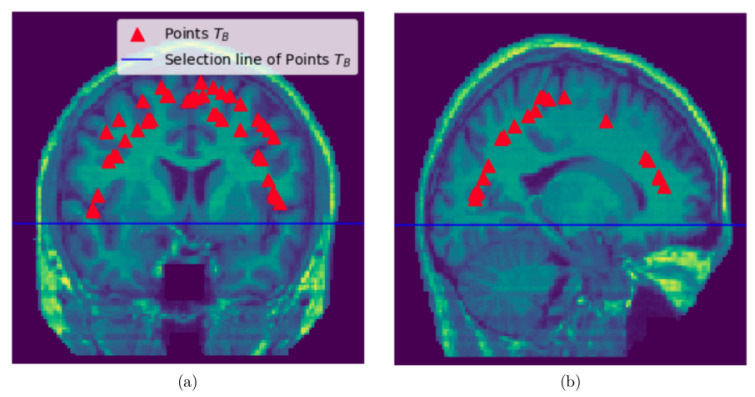
Sampling points $T_{B}$ within the brain: (**a**) Sampled slice coronal section showing the selection line and points sampled within the brain, (**b**) Sampled slice sagittal section section showing the selection line and points sampled within the brain.

**Figure 4 jimaging-08-00262-f004:**
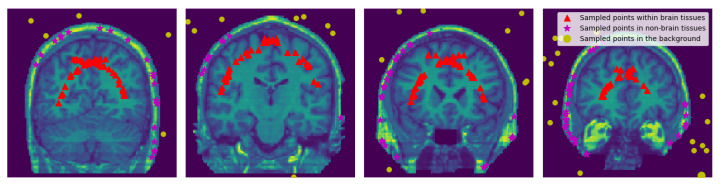
Sampled Points: 2D visualization of the representative coronal section from MRI image of a single subject (OASIS data) showing the sampled points within the brain, within the non-brain, and in the background region. $\delta_{1}$ and $\delta_{2}$ values were set to 15 and 3, respectively.

**Figure 5 jimaging-08-00262-f005:**
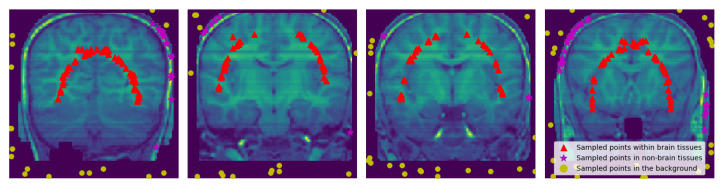
Sampled Points: 2D visualization of the representative coronal section from MRI image of a single subject (IBSR data) showing the sampled points within the brain region, within non-brain tissues, and in the background. $\delta_{1}$ and $\delta_{2}$ values were set to 15 and 3, respectively. Notice the removed part of the skull and the brain from some slices.

**Figure 6 jimaging-08-00262-f006:**
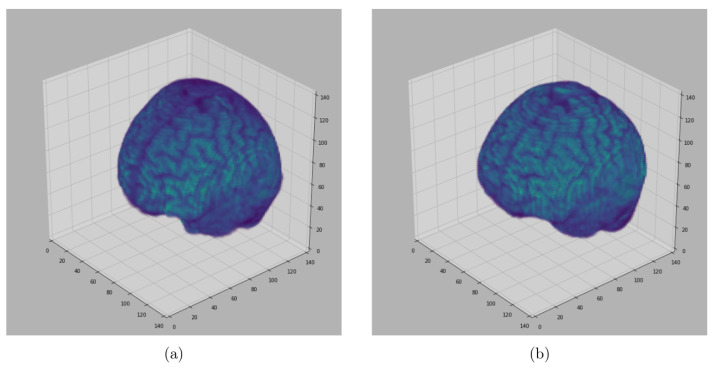
Segmented Brain (OASIS data): One representative subject representing (**a**) 3D brain segmented using GUBS approach (predicted), (**b**) 3D brain (ground truth). The masks were segmented using a custom method based on registration to an atlas, and then revised by human experts.

**Figure 7 jimaging-08-00262-f007:**
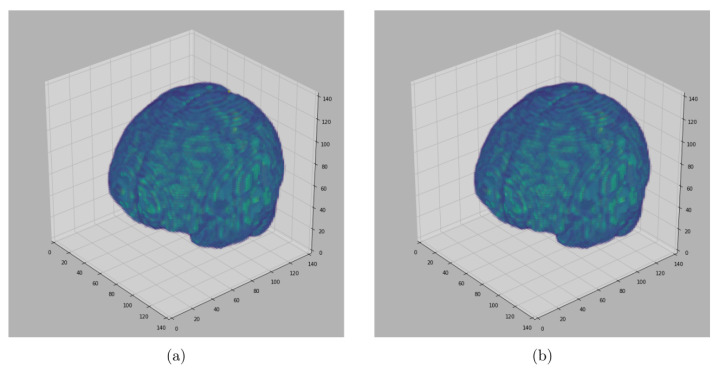
Segmented Brain (BW data): One representative subject representing (**a**) 3D brain segmented using GUBS approach (predicted), (**b**) 3D brain (ground truth). The ground truth was obtained from the labels representing cerebrospinal fluid (CSF), gray matter (GM), and white matter (WM).

**Figure 8 jimaging-08-00262-f008:**
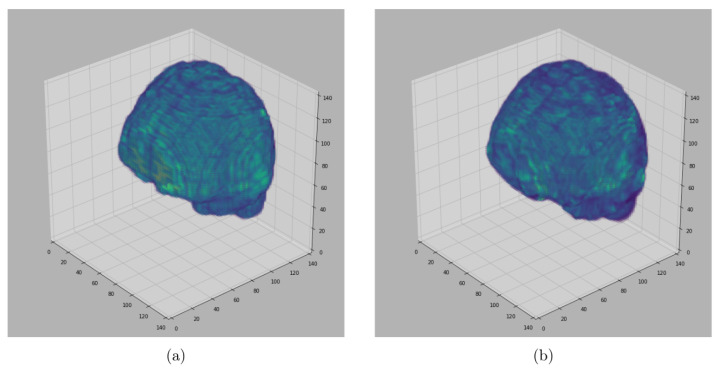
Segmented Brain (IBSR data): One representative subject representing (**a**) 3D brain segmented using GUBS approach (predicted), (**b**) 3D brain (ground truth). The ground truth was obtained by manual-guided expert segmentation.

**Figure 9 jimaging-08-00262-f009:**
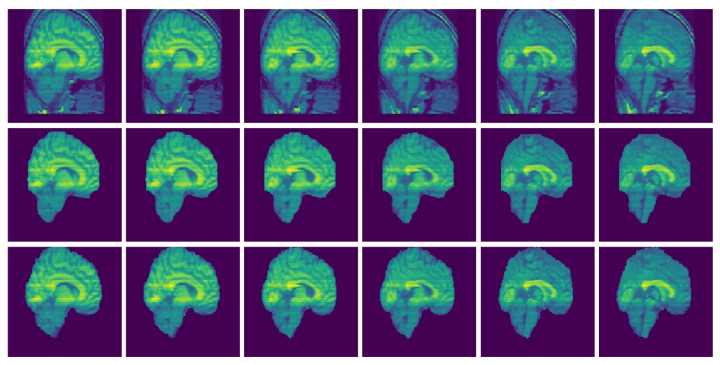
Selected MRI slices (IBSR data set): Sagittal MRI plane segmented brain. Row one: Input images, Row two: Predicted brain and Row three: Ground truth brain.

**Figure 10 jimaging-08-00262-f010:**
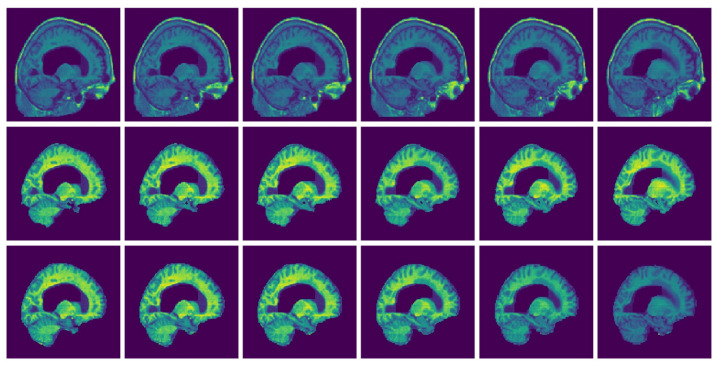
Selected MRI slices (OASIS data set): Sagittal MRI plane segmented brain. Row one: Input images, Row two: Predicted brain and Row three: Ground truth brain.

**Figure 11 jimaging-08-00262-f011:**
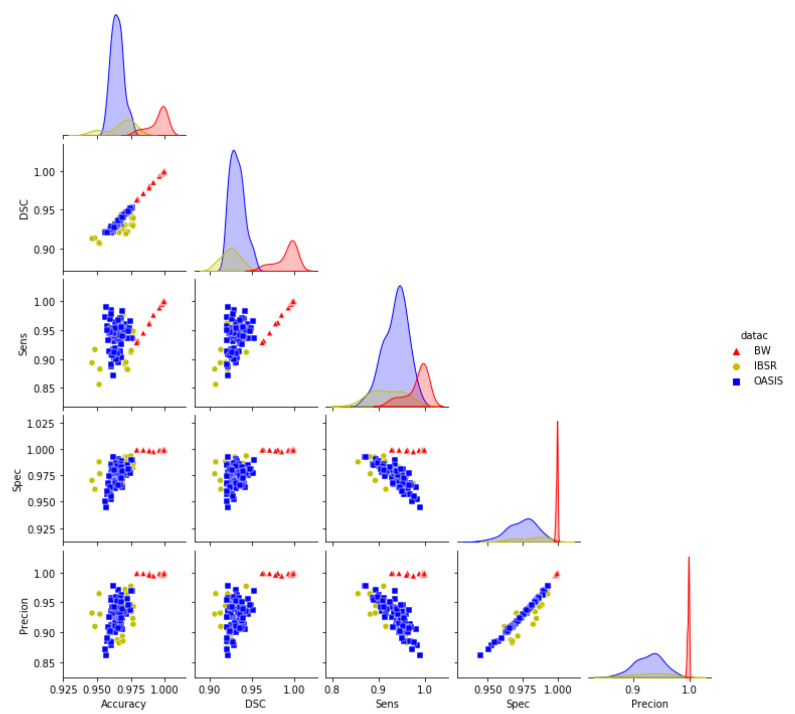
Pair plots for the measures of similarity (Combined data sets): Pair plots showing the pairwise relationship between different measures of similarity for the results obtained using GUBS across the combined data sets. DSC = Dice Similarity coefficients, Sens = Sensitivity, Spec = Specificity.

**Figure 12 jimaging-08-00262-f012:**
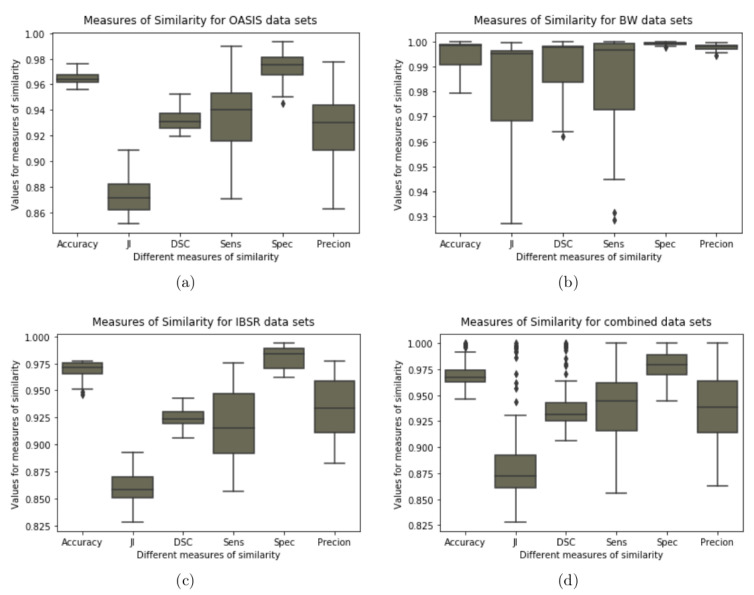
Boxplot for the measures of similarity (Combined Data sets): Boxplot showing variability for the measures of similarity for the results obtained using GUBS method across the combined data sets. JI = Jaccard Indices, DSC = Dice Similarity coefficients, Sens = Sensitivity, Spec = Specificity.

**Figure 13 jimaging-08-00262-f013:**
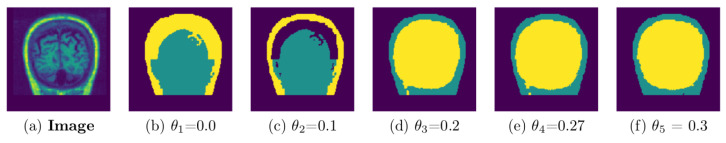
A representative coronal section from 3D MRI showing the separation of components for different thresholds: MRI from a single subject (IBSR data set) experimented using different thresholds and nodes sample size is 20,000. GUBS is run on 3D and for visualization we take 2D at the same location for all experiments.

**Figure 14 jimaging-08-00262-f014:**
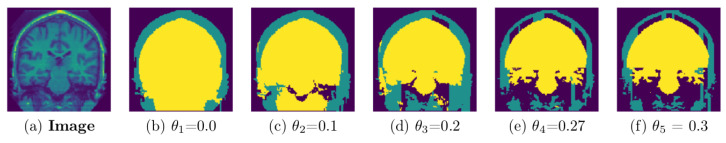
A representative coronal section from 3D MRI showing the separation of components for different thresholds: MRI from a single subject (OASIS data set) experimented using different thresholds and the nodes sample size is 20,000. GUBS is run on 3D and for visualization we take 2D at the same location for all experiments.

**Figure 15 jimaging-08-00262-f015:**
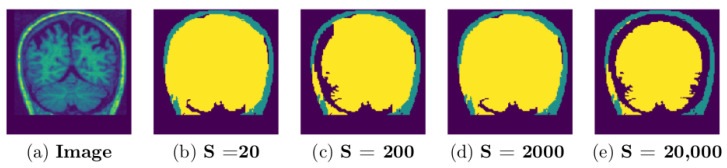
A representative coronal section from 3D MRI showing the separation of components for different sample size: MRI from a single subject (IBSR data set) experimented using threshold T=0.27 and different size of the sampled nodes. GUBS is run on 3D and for visualization we take 2D at the same location for all experiments.

**Figure 16 jimaging-08-00262-f016:**
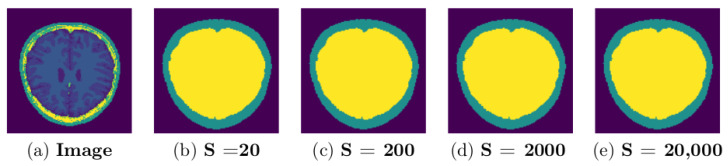
A representative axial section from 3D MRI showing the separation of components for different sample size: MRI from a single subject (BW data set) experimented without threshold, different size of the sampled nodes. GUBS is run on 3D and for visualization we take 2D at the same location for all experiments.

**Table 1 jimaging-08-00262-t001:** Performance analysis of GUBS compared to STAPLE and CONSNet [9] methods based on the OASIS data set by presenting the average in each measure of similarity.

	JI (mean ± sd)	DSC (mean ± sd)	Sensitivity (mean ± sd)	Specificity (mean ± sd)
STAPLE	-	**0.960960** ± **0.0070**	0.989830 ± 0.0060	0.951880 ± 0.0200
CONSNet	-	0.955480 ± 0.0100	**0.990550** ± **0.0060**	0.939800 ± 0.0280
GUBS	0.872633 ± 0.0148	0.931918 ± 0.0084	0.937179 ± 0.0256	**0.974223** ± **0.0101**

**Table 2 jimaging-08-00262-t002:** Performance analysis of GUBS compared to BSE, HWA and SMHASS [17] methods based on the BW data by presenting the average in each measure of similarity.

	JI (mean ± sd)	DSC (mean ± sd)	Sensitivity (mean ± sd)	Specificity (mean ± sd)
BSE	0.875000 ± 0.0490	0.932000 ± 0.0310	0.991000 ± 0.0040	0.979000 ± 0.0120
HWA	0.685000 ± 0.0170	0.813000 ± 0.0120	**1.000000** ± **0.0010**	0.928000 ± 0.0050
SMHASS	0.904000 ± 0.0110	0.950000 ± 0.0060	0.990000 ± 0.0030	0.985000 ± 0.0020
GUBS	**0.982396** ± **0.0271**	**0.990927** ± **0.0141**	0.984012 ± 0.0268	**0.999356** ± **0.0005**

**Table 3 jimaging-08-00262-t003:** Performance analysis of GUBS compared to HWA, SMHASS [17] and multi-view U-Net (MVU-Net) [19] methods based on the IBSR data set by presenting the average in each measure of similarity.

	JI (mean ± sd)	DSC (mean ± sd)	Sensitivity (mean ± sd)	Specificity (mean ± sd)
HWA	0.814000 ± 0.0360	0.897000 ± 0.0220	**1.000000** ± **0.0000**	0.966000 ± 0.0120
SMHASS	**0.905000** ± **0.0300**	**0.950000** ± **0.0170**	0.992000 ± 0.0100	0.985000 ± 0.0090
MVU-Net	-	0.908100	0.941400	**0.989400**
GUBS	0.859300 ±0.0176	0.924229 ± 0.0102	0.918936 ± 0.0334	0.980869 ± 0.0104

## Data Availability

The first data set is from the Open Access Series of Imaging Studies (OASIS) [22], www.oasis-brains.org (accessed on 15 January 2022). The second data set is from the BrainWeb (BW) database [23,24], https://brainweb.bic.mni.mcgill.ca/ (accessed on 15 January 2022). The third data set is from the Internet Brain Segmentation Repository (IBSR) [25], https://www.nitrc.org/projects/ibsr (accessed on 15 January 2022). Codes for producing results are available here https://github.com/simeonmayala/GUBS-Brain-segmentation.

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
