# Peer review of "GUBS: Graph-Based Unsupervised Brain Segmentation in MRI Images"

_2313-433X, 2022, doi:10.3390/jimaging8100262_

Round 1

Reviewer 1 Report

This work presents an updated version of Brain Segmentation in MRI Images based on existing solutions.

The authors first draw a complete state of the art for this specific problem, covering the different types of approaches. They finish with the graph-based approach on which their solution is built (ref [21]).

Then the data and the proposed approach are presented in section 2 with several interesting illustrations. Section 3 contains qualitative results with result samples on different datasets and quantitative results comparing several SOTA approaches on the same datasets. The proposed GUBS approach obtains comparable results with SOTA, better in some situations.

Finally, the Discussion and Conclusion sections summarize the work and contribution.

The paper is well written and easy to read. I list a few typos at the end of this review.

My main concern is that some parts of the explanation are missing and the authors should better explain the difference from their previous work [34]. These two points are linked because the main skip is done in section 2.2.6 where the authors refer to [34]. Using Fig 2, steps 1, 2, and the end of 4 are well described, but step 3 and the “Update the MST” in step 4 are not enough detailed in the paper for a reader who is not an expert on MST usage. In more detail, section 2.2.3 well explains how the sampling points are selected over the 3 regions, section 2.2.4 define the graph of voxels and 2.2.5 explains the principle of graph simplification by collapsing, and section 2.2.6, which is the key contribution of this paper, is not enough detailed. I think that all information is present (use the sampled points to simplify the graph before computing the MST) but: why this is a good idea? What is the difference with [34]? Why 3 sets of seeds and not only 1 ? Can you illustrate this process with a toy example? These updates will increase the originality/novelty evaluation.

Here are some other improvements following the paper order.

  • In section 2.2.1, equation 2: why consider only the dimension M to compute the minimum maximum values? it is possible to have only background voxels if you use only one row on voxels and min would be equal to the max...

  • In section 2.2.1, about the preprocessing using the threshold to clean the background: how are these thresholds obtained? on which data it has been optimized? with which criteria? If these values are shared in SOTA, add a reference.

  • In Section 2.2.2: From eq 1, A is not a binary image... so ε will not be equal to -1, 0, or 1. Please define B as the binary image of A and explain how the binarization is done (a single threshold at 0.5?)

  • In section 2.2.3, line 185: “We find points at the six surfaces of the 3D cube of the binary image”, but as shown in figures 4 and 5, the BG points are not on the faces of the cube.

  • The Discussion section is more a summary of the paper than a discussion. This section should be revised to give more space for section 2.2.6.

Some typos : 

  • line 152: there is a space before the point

  • line 153: 3D edge => 3D edges

  • line 217: a missing point before “The modified”

  • in Caption of Fig 13,14 and 15: “all experiments” with an s

Author Response

Please see the attachment. We have also included a separate .pdf file with an illustrations using a simple coins image. It is remained  Experimentation.pdf in the MDPI_template folder.

Reviewer 2 Report

The paper is well written, contains fair scientific contributions, and experiments are clearly presented. 

I have only a few minor suggestions that should be addressed in the revised version:

Line 46: It is not common to start a sentence with a reference number, please use: "Authors in [11]" or something similar.

Authors should add a GitHub link to the source code of the proposed method. It would be beneficial for the scientific community to provide source code to alleviate further research and reproducibility of the proposed method. 

Round 2

Reviewer 1 Report

I want to thank the authors for the clear response and the proposed updates. The new version of the paper is good enough for publication.